# Species Richness and Taxonomic Distinctness of Zooplankton in Ponds and Small Lakes from Albania and North Macedonia: The Role of Bioclimatic Factors

**Giorgio Mancinelli** [1,2,3], **Sotir Mali** [4] **and Genuario Belmonte** [1,5,*]

1   CoNISMa, Consorzio Nazionale Interuniversitario per le Scienze del Mare, 00196 Roma, Italy;
    giorgio.mancinelli@unisalento.it
2   Laboratory of Ecology, Department of Biological and Environmental Sciences and Technologies (DiSTeBA),
    University of Salento, 73100 Lecce, Italy
3   National Research Council (CNR), Institute of Biological Resources and Marine Biotechnologies (IRBIM),
    08040 Lesina, Italy
4   Department of Biology, Faculty of Natural Sciences, "Aleksandër Xhuvani" University, 3001 Elbasan,
    Albania; sotirmali@hotmail.com
5   Laboratory of Zoogegraphy and Fauna, Department of Biological and Environmental Sciences and
    Technologies (DiSTeBA), University of Salento, 73100 Lecce, Italy
*   Correspondence: genuario.belmonte@unisalento.it

**Abstract:** Resolving the contribution to biodiversity patterns of regional-scale environmental drivers is, to date, essential in the implementation of effective conservation strategies. Here, we assessed the species richness S and taxonomic distinctness Δ+ (used a proxy of phylogenetic diversity) of crustacean zooplankton assemblages from 40 ponds and small lakes located in Albania and North Macedonia and tested whether they could be predicted by waterbodies' landscape characteristics (area, perimeter, and altitude), together with local bioclimatic conditions that were derived from Wordclim and MODIS databases. The results showed that a minimum adequate model, including the positive effects of non-arboreal vegetation cover and temperature seasonality, together with the negative influence of the mean temperature of the wettest quarter, effectively predicted assemblages' variation in species richness. In contrast, taxonomic distinctness did not predictably respond to landscape or bioclimatic factors. Noticeably, waterbodies' area showed a generally low prediction power for both S and Δ+. Additionally, an in-depth analysis of assemblages' species composition indicated the occurrence of two distinct groups of waterbodies characterized by different species and different precipitation and temperature regimes. Our findings indicated that the classical species-area relationship hypothesis is inadequate in explaining the diversity of crustacean zooplankton assemblages characterizing the waterbodies under analysis. In contrast, local bioclimatic factors might affect the species richness and composition, but not their phylogenetic diversity, the latter likely to be influenced by long-term adaptation mechanisms.

**Keywords:** crustacean zooplankton; species richness; phylogenetic diversity; bioclimate; freshwater ponds

## 1. Introduction

Lentic freshwaters are acknowledged to play a crucial role in regulating the global ecosystem functions e.g., carbon cycle [1] and they are among the Earth's most threatened habitats in terms of intensity of anthropogenic pressures, biodiversity loss, and non-indigenous species introduction [2–4].

They include an extreme variety of habitats differing in ecological characteristics and fragility [5,6]. Surface area represents one of the most apparent differentiating properties: indeed, lentic environments (304 million water bodies; 4.2 million $km^2$ in total area [7]) include the lake Superior (82,000 $km^2$), together with small ponds, i.e., waterbodies less than 0.05 $km^2$ in area [8].

Ponds and small lakes (hereafter PSL) have significant ecological functions [9,10]: among others, they provide a considerable contribution to inland water $CO_2$ and $CH_4$ emissions [11]. In addition, they are, to date, recognized as important biodiversity hotspots, especially in mountainous regions, supporting a high species richness and contributing a high degree of rare species to regional pools ([12–15]; see also [16] for an example on planktonic Calanoida). Noticeably, PSL are threatened by a number of anthropogenic pressures, including nutrient loading, contamination, acid rain, and invasion of exotic species [17]. In addition, infilling (both natural and caused by direct habitat destruction), land drainage, decline in many of their traditional uses, and changes of function determine at a regional scale the drastic reduction in PSL number and connectivity [12].

In the last decade, several investigations have focused on the diversity of benthic invertebrates, as they are excellent bio-indicators of PSL ecological integrity [18,19]. Local factors that are related with e.g., hydroperiod, environmental harshness, water chemistry, spatial connectivity, habitat heterogeneity, and presence of predators, have been recognized to influence the biodiversity of macroinvertebrate assemblages ([20] and literature cited). At a regional scale, attention has been primarily given to the influence of waterbodies area [21–23], while assuming, within the general theoretical background provided by the species-area relationship (SAR) hypothesis [24], that basins' size correlates with the number of microhabitats within the basin itself and with populations' abundance, and thence inversely correlated with the likelihood of random extinctions. However, resolving the contribution to biodiversity patterns of environmental factors acting at a regional scale is, to date, essential to the implementation of effective conservation strategies in the face of e.g. deforestation and climate change ([25,26] and literature cited). Accordingly, several attempts have been made to model biodiversity of freshwater environments by means of regional bioclimatic factors [27,28].

In the present study, we focused on the diversity of crustacean zooplankton assemblages in 40 ponds and small lakes differing remarkably in terms of origin, extension, and altitude from a relatively wide region comprising part of Albania and North Macedonia. A recent faunal inventory focusing on ponds and lakes in the area [29] provided the starting reference information on the taxonomic characteristics of the assemblages.

A number of studies have generally indicated a positive relationship between the surface area of lacustrine environments and zooplankton diversity (e.g., [30–32]; but see [13]); accordingly, crustacean zooplankton has been shown to have higher species richness in small ponds as compared with lakes [16,33,34]. This notwithstanding, we hypothesized that area alone may not be an adequate predictor, and that local bioclimatic conditions may ultimately contribute in explaining diversity variations across waterbodies by influencing their physical-chemical characteristics, as observed in recent investigations on freshwater macroinvertebrates and macrophytes [27,28,35]. This could be particularly true for waterbodies in mountainous habitats, where temperature and precipitation regimes intensely reflect the chemical-physical characteristics and hydroperiod of the waterbodies themselves [36], regulating the harshness and stability of the aquatic environments and, in turn, the diversity of the biota living in them ([22] and literature cited).

To verify the hypothesis and test whether bioclimatic factors can predict assemblages' diversity, we identified a minimum adequate model (MAM) predicting assemblages' diversity across the different waterbodies by means of a heuristic multiple regression approach and Bayesian Information Criterion model selection method while using satellite-derived bioclimatic variables as predictors. Multiple indices are, to date, available to quantify different aspects of biodiversity [37]. Here, we identified predictive MAMs estimating the diversity of crustacean zooplankton assemblages in terms of species richness and average taxonomic distinctness.

Species richness is the most classical measure of biodiversity across ecosystems that has been extensively used in studies on lentic habitats (see references cited above). This index provides an incomplete understanding of biological variability, because it neglects information on the identity and taxonomic relationship among species, and it is hampered by a number of critical limitations [38,39]. Accordingly, we used the average taxonomic distinctness Δ+ [40] to compare the taxonomic relatedness of species in the crustacean assemblages of every water body. In addition, we tested the influence of bioclimatic factors on crustacean assemblages in terms of species composition. To this end, multivariate approaches that are based on a canonical analysis of principal coordinates were used to model the changes in the structure of the assemblages as affected by bioclimatic variables, and identify relationships between the latter and specific groups of zooplankton taxa.

## 2. Material and Methods

### 2.1. Sampling Sites and Collection Procedures

A total of 40 sites were selected among those (53) that were surveyed between 2005 and 2017 by Belmonte and colleagues [29] in an area comprised between 39°55′22″–42°04′30″ N, and 19°24′30″–20°47′36″ E. (Figure 1, Table 1).

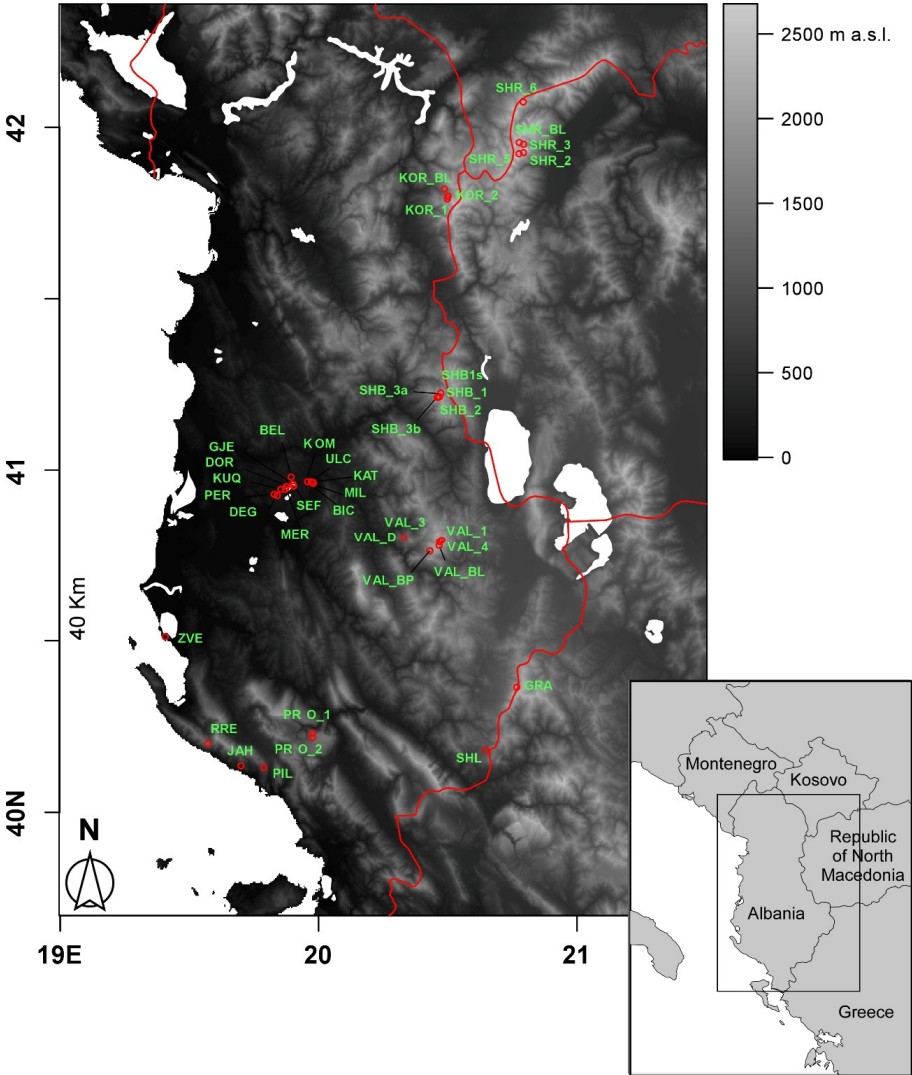

**Figure 1.** Location of the 40 waterbodies in Albania and North Macedonia.

**Table 1.** List of the waterbodies included in the study. The code used in Figure 1 is included, together with information of the basins origin (N = natural, A = artificial), typology (L= lake, P = pond (area < 0.05 km$^2$)), elevation (in m), coordinates, number of samples collected, year of collection, species richness S, and taxonomic distinctness Δ+.

| Waterbody | Code | Origin | Typology | Altitude | Latitude | Longitude | Samples | Year | S | Δ+ |
|---|---|---|---|---|---|---|---|---|---|---|
| Bellsh | BEL | N | L | 150 | 40°58′47″ | 19°53′37″ | 2 * | 2008 | 6 | 76.67 |
| Bici | BIC | N | P | 77 | 40°57′41″ | 19°58′46″ | 2 * | 2009 | 8 | 74.11 |
| Dega | DEG | N | L | 105 | 40°55′32″ | 19°50′25″ | 2 * | 2009 | 8 | 60.27 |
| Dorbi | DOR | N | L | 133 | 40°57′07″ | 19°52′30″ | 3 * | 2008 | 9 | 69.1 |
| Gjeluar | GJE | N | P | 127 | 40°57′32″ | 19°54′02″ | 3 * | 2010 | 8 | 74.11 |
| Gramoz | GRA | N | P | 2364 | 40°21′52″ | 20°47′26" | 2 | 2008 | 4 | 81.25 |
| South Coast Jahl | JAH | A | P | 701 | 40°07′53″ | 19°47′16″ | 2 | 2012 | 3 | 75 |
| Katund | KAT | N | P | 100 | 40°57′54″ | 19°58′27″ | 2 * | 2009 | 8 | 75.45 |
| Komnjec | KOM | N | P | 135 | 40°57′58″ | 19°57′26″ | 3 * | 2009 | 5 | 80 |
| Korab Hapave 1 | KOR1 | N | P | 1786 | 41°47′59″ | 20°30′04″ | 1 | 2017 | 14 | 67.86 |
| Korab Hapave 2 | KOR2 | N | P | 1779 | 41°47′55″ | 20°30′03″ | 1 | 2017 | 11 | 70.68 |
| Korab black lake | KOR BL | N | P | 1470 | 41°49′13″ | 20°29′14″ | 1 | 2017 | 7 | 69.05 |
| I Kuq | KUQ | N | L | 135 | 40°56′38″ | 19°51′14″ | 3 * | 2008 | 8 | 75 |
| Merohjes | MER | N | L | 112 | 40°56′39″ | 19°52′19″ | 2 * | 2008 | 11 | 72.27 |
| Milosh | MIL | N | L | 70 | 40°57′49″ | 19°58′50″ | 2 * | 2009 | 8 | 65.63 |
| Pernaska | PER | N | L | 107 | 40°55′45″ | 19°49′40″ | 3 * | 2008 | 6 | 80.83 |
| South Coast Pilur | PIL | A | P | 280 | 40°08′08″ | 19°41′59″ | 1 | 2012 | 3 | 75 |
| Progonat 1 | PRO1 | A | P | 1325 | 40°13′06″ | 19°58′35″ | 1 | 2011 | 4 | 66.67 |
| Progonat 2 | PRO2 | A | P | 1262 | 40°13′50″ | 19°58′35″ | 1 | 2011 | 4 | 72.92 |
| Karaburun Rreza | RRE | A | P | 1333 | 40°12′03″ | 19°34′17″ | 2 | 2012 | 3 | 79.17 |
| Seferan | SEF | N | L | 124 | 40°57′12″ | 19°54′12″ | 1 * | 2011 | 9 | 71.53 |
| Shebenik 1 | SHB1 | N | P | 1903 | 41°13′30″ | 20°28′21″ | 1 | 2015 | 6 | 75.83 |
| Shebenik 2 | SHB2 | N | P | 2006 | 41°12′50″ | 20°28′04″ | 1 | 2015 | 5 | 60 |
| Shebenik 3a | SHB3a | N | P | 2054 | 41°12′44″ | 20°27′43″ | 1 | 2015 | 4 | 64.58 |
| Shebenik 3b | SHB3b | N | P | 2005 | 41°12′45″ | 20°27′31″ | 1 | 2015 | 6 | 75.83 |
| Shebenik 1s | SHB1s | N | P | 1905 | 41°13′29″ | 20°28′21″ | 1 | 2015 | 5 | 78.75 |
| Sheleguri | SHL | A | L | 1002 | 40°10′55″ | 20°38′49″ | 1 | 2012 | 4 | 72.92 |
| Sharr 2 | SHR2 | N | P [#] | 2280 | 41°57′21″ | 20°46′34″ | 2 | 2008 | 5 | 76.25 |
| Sharr 3 | SHR3 | N | L | 1945 | 41°57′02″ | 20°47′36″ | 2 | 2008 | 11 | 68.41 |
| Sharr 5 | SHR5 | N | P | 2435 | 41°55′23″ | 20°46′32″ | 1 | 2016 | 4 | 68.75 |
| Sharr 6 | SHR6 | N | P | 2190 | 42°04′26″ | 20°47′32″ | 2 | 2016 | 8 | 71.43 |

**Table 1.** *Cont.*

| Waterbody | Code | Origin | Typology | Altitude | Latitude | Longitude | Samples | Year | S | Δ+ |
|---|---|---|---|---|---|---|---|---|---|---|
| Sharr black lake | SHR BL | N | P # | 2170 | 41°55′34″ | 20°47′34″ | 2 | 2016 | 7 | 66.67 |
| Ulca | ULC | N | L | 107 | 40°57′57″ | 19°58′17″ | 2 * | 2009 | 7 | 71.43 |
| Valamare 1 | VAL1 | N | P | 2051 | 40°47′40″ | 20°28′36″ | 1 | 2016 | 7 | 73.81 |
| Valamare 3 | VAL3 | N | P | 2062 | 40°47′38″ | 20°28′29″ | 1 | 2016 | 8 | 70.98 |
| Valamare 4 | VAL4 | N | P | 2121 | 40°47′19″ | 20°28′05″ | 1 | 2016 | 9 | 71.53 |
| Valamare black lake | VAL BL | N | P | 1698 | 40°45′43″ | 20°25′50″ | 1 | 2016 | 7 | 73.21 |
| Valamare green bun pine | VAL BP | N | P | 2005 | 40°46′40″ | 20°28′00″ | 1 | 2016 | 8 | 69.2 |
| Valamare Dushq teke | VAL D | N | L | 1115 | 40°48′06″ | 20°19′47″ | 2 | 2010 | 6 | 70 |
| Narte Zvernec | ZVE | N | P | 2 | 40°30′41″ | 19°24′24″ | 1 | 2012 | 6 | 71.67 |

# temporary pond. * sample collection performed while using a canoe.

Sampling procedures are described in detail elsewhere [29]. As the water bodies varied in terms of area, altitude, origin (natural, artificial), as well as in hydrology (permanent, temporary), banks morphology, and degree of aquatic vegetation cover, it was not possible to apply a standardized sampling protocol across all of the sites.

In the selected sites, sampling operations were carried out in spring–early summer (i.e., between April and July) always by the same operators, while using a hand-held plankton net (200 μm mesh-size, mouth diameter, 30 cm). The collection (by plankton net towing from two opposite edges of the pond) covered the whole water body when its diameter was smaller than 100 m. For larger water bodies, sample collection was carried while using a canoe. In the case of small ponds, the collection procedure was repeated three times (each sample derived from the execution of three collections). In the case of larger water bodies, a sample collection was carried out in three different stations; the three different samples were ultimately cumulated.

After collection, the samples were fixed in situ in 90–96% ethanol. In the laboratory, taxa were identified to the species level while using a compound microscope (×30–×300 magnifications) that was equipped with a *camera lucida*.

The quantification of the abundance of each taxon was not performed, and only presence/absence data were considered due to the huge variability of water volumes in each pond, which made impossible the comparison among the concentrations of plankton of different sites.

## 2.2. Landscape-Climate Variables

Together with altitude, we used area and perimeter of the water bodies together with bioclimatic factors (i.e., temperature and precipitation) to represent the landscape-climate variables. Water bodies were geo-referenced in Google Earth Pro version 7.3.2., where their surface (in $km^2$) and perimeter (in km) were measured while using the software tools. Measurements were performed by preferentially choosing images that were taken in spring or summer between 2008 and 2017, assuming that negligible variations in the water bodies size and morphology occurred during this period.

Nineteen climatic layers with a 30-second spatial resolution ($0.93 \times 0.93 = 0.86$ $km^2$ at the equator; approximately $0.92 \times 0.70 = 0.64$ $km^2$ within the study area), including temperature and precipitation variables, were extracted from the WorldClim v2 data set [41] (Table A1 in online material). Besides climate, there is a growing recognition of the importance of vegetation cover in characterizing the spatial environmental heterogeneity in a given area at the meso- and topo-scales, in turn affecting climate, soil composition, hydrology and geomorphology, and, ultimately, biological processes that were related with species richness and community complexity [42,43]. Accordingly, two vegetation variables (i.e., percent tree cover and percent non-tree cover) were extracted from the Terra MODIS Vegetation Continuous Field (VCF) product (available as MOD44B v006 https://lpdaac.usgs.gov/products/mod44bv006/). Percent tree cover included all forest types and age classes, while percent non-tree cover included meadows, regeneration areas, and clear-cut areas. MODIS tiles of the study area were re-projected and re-sampled to meet the coordinate system and resolution of WorldClim layers; percent cover data were subsequently obtained for the years from 2006 to 2017, and averaged. In addition, the third VCF component of ground cover, i.e., percent bare soil (including bare soils and rocks) was extracted according to the aforementioned procedures, and was used together with tree and non-tree vegetation percent cover data to estimate the Shannon's diversity index (H) as a proxy of habitat heterogeneity.

## 2.3. Data Analysis

The values in the text are expressed as averages ± 1SE; for parametric statistical analysis, data were tested for conformity to assumptions of variance homogeneity (Cochran's C test) and normality (Shapiro–Wilks test) and transformed when required.

The taxonomic diversity of crustacean assemblages in each water body was estimated in terms of species richness S and average taxonomic distinctness Δ+. The index can be used as a proxy for

phylogenetic diversity and it measures the mean path length through a taxonomic tree connecting every species [40]. Here, mean taxonomic distinctness values were calculated assigning equal weighting to branch lengths from a linear Linnaean classification while using eight taxonomic levels (i.e., class, subclass, order, suborder, infraorder, family, genus, and species). The taxonomic classification tree was built according with the World Register of Marine Species (WoRMS, available at https://www.marinespecies.org) and the Integrated Taxonomic Information System (ITIS, available at https://www.itis.gov).

For the sake of completeness, other taxonomic diversity indices were calculated, including the total taxonomic distinctness $s\Delta+$ and the variance in taxonomic distinctness $\Lambda+$ [44,45], the average phylogenetic diversity $\Phi+$, and the total phylogenetic diversity $s\Phi+$ [46]. S resulted in being significantly related with $s\Delta+$, $\Phi+$, and $s\Phi+$ ($r = 0.98$, $-0.89$, and $0.95$, respectively; $P$ always $< 0.01$, 38 degrees of freedom), while $\Delta+$ scaled negatively with $\Lambda+$ ($r = -0.44$, $P = 0.004$); conversely, S and $\Delta+$ were characterized by a non significant negative correlation ($r = -0.26$, $P = 0.12$, 38 d.f.); thus, the two indices were chosen for further analyses.

We verified the influence on both indices of potential artefacts, due to (i) possible differences in sampling procedures and (ii) the different number of total collected samples per water body. To this end, we performed a one way permutational analysis of variance (PERMANOVA; [47]) based on Euclidean distances and 999 permutations with "sampling procedure" as a fixed factor (two levels, "hand", or "canoe") and the number of collected samples as the covariate (P and N hereafter). Both factors exerted negligible influences on the S and $\Delta+$ estimations (S: Pseudo–$F_P = 4.42$, P(perm)$_P = 0.08$, Pseudo–$F_N = 0.48$, P(perm)$_N = 0.47$, Pseudo-$F_{P \times N} = 0.12$, P(perm)$_{P \times N} = 0.72$; $\Delta+$: Pseudo–$F_P = 1.67$, P(perm)$_P = 0.21$, Pseudo–$F_N = 0.01$, P(perm)$_N = 0.94$, Pseudo–$F_{P \times N} = 1.07$, P(perm)$_{P \times N} = 0.31$). Consequently, the S and $\Delta+$ values were assumed to provide robust estimation of planktonic Crustacea diversity across the studied water bodies. PERMANOVA was further used to test the effects of the factors "origin" (two levels, "natural" and "artificial") and "typology" (two levels, "pond" and "lake") on the diversity indices. As water bodies varied greatly in elevation (Table 1), the latter was included in the analyses after log-transformation as a continuous covariate.

The georeferenced locations of the sampling sites were used to extract climatic and vegetation data from environmental layers. The final data set included 19 climatic, two vegetation (% tree cover, % non-tree cover), four geomorphological (elevation, perimeter, surface, and perimeter/surface ratio), and one habitat heterogeneity variable (Table A1). They were log-transformed and z-scaled; subsequently, their original number (25) was reduced while using an iterative variance inflation factor (VIF) analysis [48,49]. In brief, if a strong linear relationship links a variable $x$ with at least another variable $y$, the correlation coefficient would be close to 1, and the VIF for $x$ would be large. Here, diversity measures with VIF factors that were larger than 10 were excluded. Variables with VIF factors larger than 10 were discarded. The identification of a minimum adequate model (MAM hereafter; [50]) linking diversity measures with environmental variables was based on the heuristic generation of alternative regression models (see [51] for complete details on the procedure). Model selection was performed while adopting an Information Theoretic criterion [52]; the second-order Akaike Information Criterion *AICc* [53,54] was calculated for each combination of $n$ explanatory variables and used to identify the best MAM among the alternative regression models that were generated by the procedure. For model comparison, *AICc* values were used to estimate a set of positive Akaike weights $w_i$ summing 1:

$$w_i(AIC) = \frac{\exp[-1/2(AIC_i - minAIC)]}{\sum_1^K \exp[-1/2(AIC_i - minAIC)]} \tag{1}$$

With $K$ = number of models. The model showing the highest $w_i$ was accepted as the best candidate; other candidate models were accepted if characterized by $w_i$ values within 12.5% of the highest [55–57]. The model building and MAM identification procedures were performed while adopting Fox and Weisberg [58] as a general reference.

Given the non-conclusive outcomes of the analyses that were performed on taxonomic distinctness (see Results section), we verified whether bioclimatic factors influenced planktonic Crustacea assemblages in terms of species composition. Species incidence data were used to calculate a Jaccard similarity matrix across the different waterbodies. In addition, a similarity matrix that was based on Euclidean distances was constructed for bioclimatic variables, and the consistency of the two matrices was tested while using the Kendall coefficient of concordance (W). Subsequently, a canonical analysis of principal coordinates (CAP) [59] was performed to model changes in assemblages composition, as affected by bioclimatic factors. The appropriate number of principal coordinates *m* was chosen as to minimise the *P* value from the permutation test based upon the trace statistic and maximizing the leave-one-out allocation success [60]. Post-hoc PERMANOVA tests were performed to confirm the results of the ordination for both bioclimatic factors and species; SIMPER analyses were performed on the Euclidean distance matrix of the former to assess the percentage contribution of each factor to the dissimilarity between the groups of waterbodies that were identified by the CAP procedure. Furthermore, the Spearman's rank correlations were estimated to identify the bioclimatic variables and the crustacean species that most effectively described the groups of waterbodies that were identified by the CAP procedure. Only the variables with a Spearman rank correlation coefficient *r* > 0.55 were considered.

All of the analyses were implemented in the R statistical environment v3.6.1 [61] while using a suite of packages including *taxize* (for taxonomic information retrieval from online databases) *vegan* (for diversity measures and multivariate analyses), *raster*, *rgdal*, and *maptools* (for environmental layers manipulation), *usdm* (for VIF analysis), *car*, *leaps*, and *HH* (for MAM identification).

## 3. Results

### 3.1. General Features

The 40 water bodies analysed, varied remarkably in terms of altitude, area, and perimeter (Figure 2). They showed an average altitude of 1168.3 m a.s.l. (± 141.9 m SE), ranging from 2 m a.s.l. (Narte Zvernec pond, ZVE in Figure 1) to 2,435 m a.s.l. (SHR 5). The average surface extension was 0.09 km$^2$ ± 0.03 SE, ranging from $9.1\times10^{-4}$ to 0.86 km$^2$. The average perimeter was 1.01 ± 0.21 km, ranging between 0.035 and 6.1 km. For both of the variables, the minimum and maximum values corresponded with a small artificial pond in the karst highlands of Progonat (PRO1) and the Lake Seferan in the Dumre region (SEF).

In the 40 water bodies, 79 Crustacea species were identified in total, being almost equally distributed between the classes Branchiopoda and Hexanauplia (41 and 38 species respectively).

Among Branchiopoda, Anomopoda outnumbered the other two orders Ctenopoda and Haplopoda (38 vs. 2 and 1 species). *Daphnia*, *Ceriodaphnia*, and *Moina* were the genera that were characterized by the highest number of species (nine *Daphnia* species, four *Moina* species, and three *Ceriodaphnia* species) together representing the majority of all the sampled Anomopoda species. Ctenopoda were represented by the congeneric *Diaphanosoma brachyurum* and *D. lacustris* while Haplopoda by the single species *Leptodora kindtii*. The class Hexanauplia (*alias* Copepoda) was dominated by species belonging to the order Cyclopoida (31) and to a minor extent Calanoida (7). The genus *Cyclops* (seven species) together with *Acanthocyclops*, *Paracyclops* (four species each), and *Mesocyclops* (three species) constituted to the majority of the species in the order; Calanoida were represented by the genera *Eudiaptomus* (three species), *Mixodiaptomus* (two species each), and by *Arctodiaptomus salinus* and *Neodiaptomus schmackeri*.

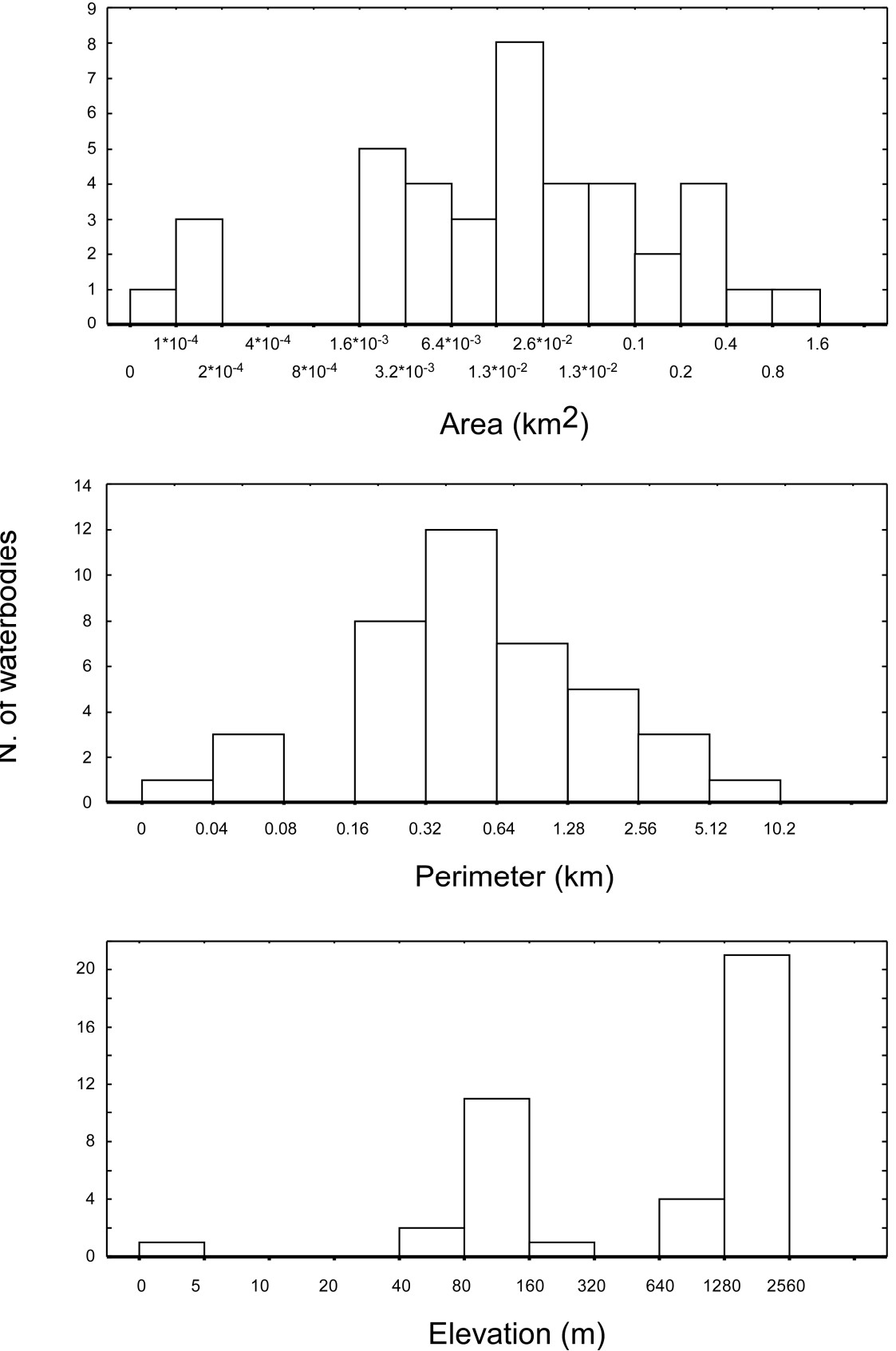

**Figure 2.** Frequency distribution of waterbodies area, perimeter, and altitude. The histograms for first variables adopt geometric scale increments (×2), while for altitude (elevation) a linear scale is used.

The species showing the widest distributions were the Anomopoda *Bosmina longirostris* (16 sites), *Chydorus sphaericus* (15 sites), and *Daphnia longispina* (14 sites) (Table 2); in addition, the Cyclopoida *Mesocyclops leuckarti*, the Calanoida *Mixodiaptomus tatricus*, and the Ctenopoda *Diaphanosoma brachyurum* occurred in 12 sampled sites.

**Table 2.** Summary of PERMANOVAs on species richness S and taxonomic distinctness + of crustacean zooplankton assemblages testing for the effects of waterbodies' origin and typology including elevation as a continuous covariate **: $p < 0.01$.

| Response Variable | S | | | Δ+ | |
|---|---|---|---|---|---|
| Source of variation | df | MS | Pseudo-F | MS | Pseudo-F |
| Altitude (1) | 1 | 1.93 | 0.41 | 9.43 | 0.35 |
| Origin (2) | 1 | 71.68 | 15.16 ** | 18.83 | 0.69 |
| Hydrology (3) | 1 | 11.19 | 2.37 | 29.73 | 1.09 |
| 1 × 2 | 1 | 0.16 | $3.4 \times 10^{-2}$ | 1.01 | $3.7 \times 10^{-2}$ |
| 1 × 3 | 1 | 0.11 | $2.2 \times 10^{-2}$ | 1.39 | $5.2 \times 10^{-2}$ |
| 2 × 3 | 1 | 0.82 | 0.17 | 1.11 | $4.1 \times 10^{-2}$ |

As regarding high level taxa, only Anomopoda (Cladocera) were present in every site. Cyclopoida (Hexanauplia) were present in 38 sites (95% of the total), Calanoida (Hexanauplia) in 30 sites (75%), Ctenopoda (Cladocera) in four sites (10%), and Haplopoda (Cladocera) in three sites (7.5%).

### 3.2. Diversity Patterns and Bioclimatic Correlates

On average, $6.7 \pm 0.4$ species per water body were found, ranging between three (JAH, PIL, RRE) and 14 species (KOR 1). The taxonomic distinctness Δ+ of the different planktonic assemblages was on average $72.1 \pm 0.78$, ranging between 60 (SHB 2) and 88.9 (GRA).

The factor "origin" was the only exerting significant effects of the species richness of waterbodies (Table 2), with the six artificial water bodies being included in the study characterized by lower S values as compared with natural basins ($3.5 \pm 0.22$ vs. $7.29 \pm 0.38$, respectively). Conversely, negligible effects were generally observed for the taxonomic distinctness Δ+ (Table 2).

The 25 predictor variables (Table A1) were reduced to a set of nine characterized by negligible collinearity (Table A2). They included five climatic variables (i.e., Isothermality, Temperature Seasonality, Mean Temperature of Wettest Quarter, Annual Precipitation, and Precipitation of Coldest Quarter), % tree and % non-tree vegetation cover, habitat heterogeneity, and water body surface. Besides water body surface, the variable Mean Temperature of Wettest Quarter showed the greatest among-water bodies variability (Table A2), ranging from a minimum of −5.32 °C (SHB 2) to a maximum of 11 °C (ZVE). It was followed by % tree cover (varying between 1 and 70%, BEL and DEG, respectively) and % non-tree cover (ranging between 20 and approx. 81%, BEL-DEG and SHR 2, respectively).

The heuristic search procedure identified a Minimum Adequate Model (MAM) predicting the variation of species richness S across the different water bodies relying on the three explanatory variables % non tree cover, Temperature Seasonality, and Mean Temperature of Wettest Quarter (Figure 3; multiple $r = 0.58$, $P = 0.002$, d.f. = 3, 36). The MAM was characterized by the lowest *AICc* value, and by an Akaike weight $w_i$ approximately eight times larger than the second-best candidate, based on the variables Temperature Seasonality, Mean Temperature of Wettest Quarter, and Habitat heterogeneity (Table 3). All of the predictors provided significant contributions to S variation across water bodies (minimum absolute *t* value = 2.34, $P = 0.02$ for the variable Mean Temperature of Wettest Quarter); the contributions of both % non-tree cover and Temperature Seasonality were positive ($b = 0.27 \pm 0.12$ and $0.64 \pm 0.15$, respectively), while the Mean Temperature of the Wettest Quarter provided a negative contribution ($b = -0.26 \pm 0.14$). Noticeably, none of the first ten best-performing models included water body area (Table 3).

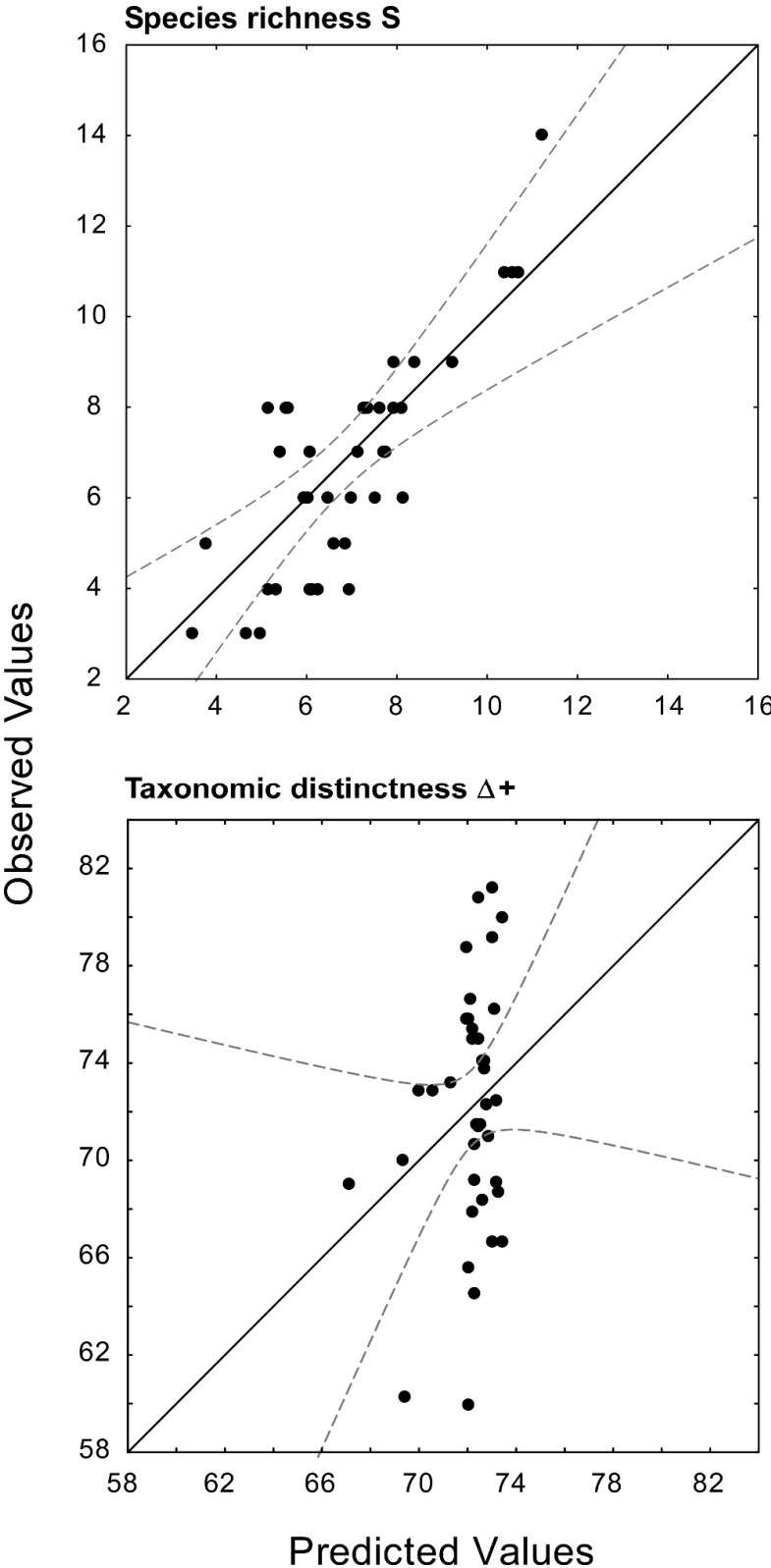

**Figure 3.** Species richness S of crustacean zooplankton assemblages in the 40 waterbodies under analysis plotted against the values predicted by the best Minimum Adequate Model (MAM; see Table 3 for predictor variables) identified adopting a multiple regression procedure (top). Dashed curves are 95% CI. For the sake of completeness the predicted taxonomic distinctness values Δ+ by the best MAM are also reported (bottom).

**Table 3.** Summary of the heuristic multiple regression analysis followed by a parsimonious selection procedure of the Minimum Adequate Model (MAM) predicting species richness (S) and of crustacean zooplankton assemblages by means of bioclimatic variables; only the first 10 best MAMs are reported. For the sake of completeness, results of MAM analysis are reported also for taxonomic distinctness ($\Delta+$), even though the statistical power of the models was negligible (see Results). K: number of predictors included in the model; AICc: second-order Akaike Information Criterion; $w_i$: Akaike weight. For predictor abbreviations see Table A1.

| | Species Richness S | | |
|---|---|---|---|
| **K** | **Predictors** | *AICc* | $w_i$ |
| 3 | %nontr–TempSea–MeanTwet | 62.84 | 0.485 |
| 3 | H–TempSea–MeanTwet | 67.05 | 0.058 |
| 3 | %nontr–TempSea–PrecColdQ | 67.07 | 0.057 |
| 4 | H–Iso–TempSea–MeanTwet | 67.12 | 0.046 |
| 2 | TempSea–PrecColdQ | 67.57 | 0.044 |
| 4 | %nontr–TempSea–MeanTwet–Prec | 67.62 | 0.044 |
| 4 | %nontr–TempSea–MeanTwet | 67.71 | 0.043 |
| 3 | TempSea–MeanTwet–Prec | 67.73 | 0.042 |
| 1 | Iso–PrecColdQ | 67.74 | 0.041 |
| 4 | %nontr–Iso–TempSea–MeanTwet | 67.96 | 0.04 |
| | Taxnomic Distinctness$\Delta+$ | | |
| **K** | **Predictors** | *AICc* | $w_i$ |
| 1 | %tr | 130.92 | 0.19 |
| 2 | %tr–MeanTwet | 131.88 | 0.12 |
| 1 | PrecColdQ | 131.98 | 0.11 |
| 2 | %tree–PrecColdQ | 132.08 | 0.11 |
| 1 | MeanTwet | 132.42 | 0.09 |
| 3 | %tr–MeanTwet–SUR | 132.58 | 0.08 |
| 2 | MeanTwet–SUR | 132.67 | 0.08 |
| 1 | %nontr | 132.71 | 0.08 |
| 1 | SUR | 132.74 | 0.08 |
| 2 | %tr–SUR | 132.79 | 0.07 |

In contrast with species richness, the heuristic search procedure was unable to identify a MAM with a significant predictive power for taxonomic distinctness. The single variable % tree cover, resulted the best predictor of $\Delta+$ (Table 3); however, the correlation resulted in being non-significant ($r = 0.25$, $P = 0.11$, d.f. 1,38; see also Figure 3). Other models showed an even worst performance, independently from the number of variables involved (Table 3), indicating, in turn, that the taxonomic distinctness of the crustacean assemblages cannot be predicted by bioclimatic, landscape-scale factors.

Canonical analysis of principal coordinates (CAP), followed by a confirmatory PERMANOVA test identified two main groups of waterbodies significantly different in terms of species composition (Figure 4; Pseudo–F = 7.2, P(perm) = 0.001). The variables Isothermality, Mean Temperature of Wettest Quarter, Annual Precipitation, and Precipitation of Coldest Quarter showed a correlation ($r > 0.65$) with the canonical axis 1 (Figure 4) and significantly differed between the two groups of waterbodies (PERMANOVA, Pseudo–F = 27.4, P(perm) = 0.001). A Simper procedure indicated that the variable Mean Temperature of Wettest Quarter contributed by 32.2% to inter-group differences, followed by Isothermality and Annual Precipitation (28.6 and 24.2%, respectively).

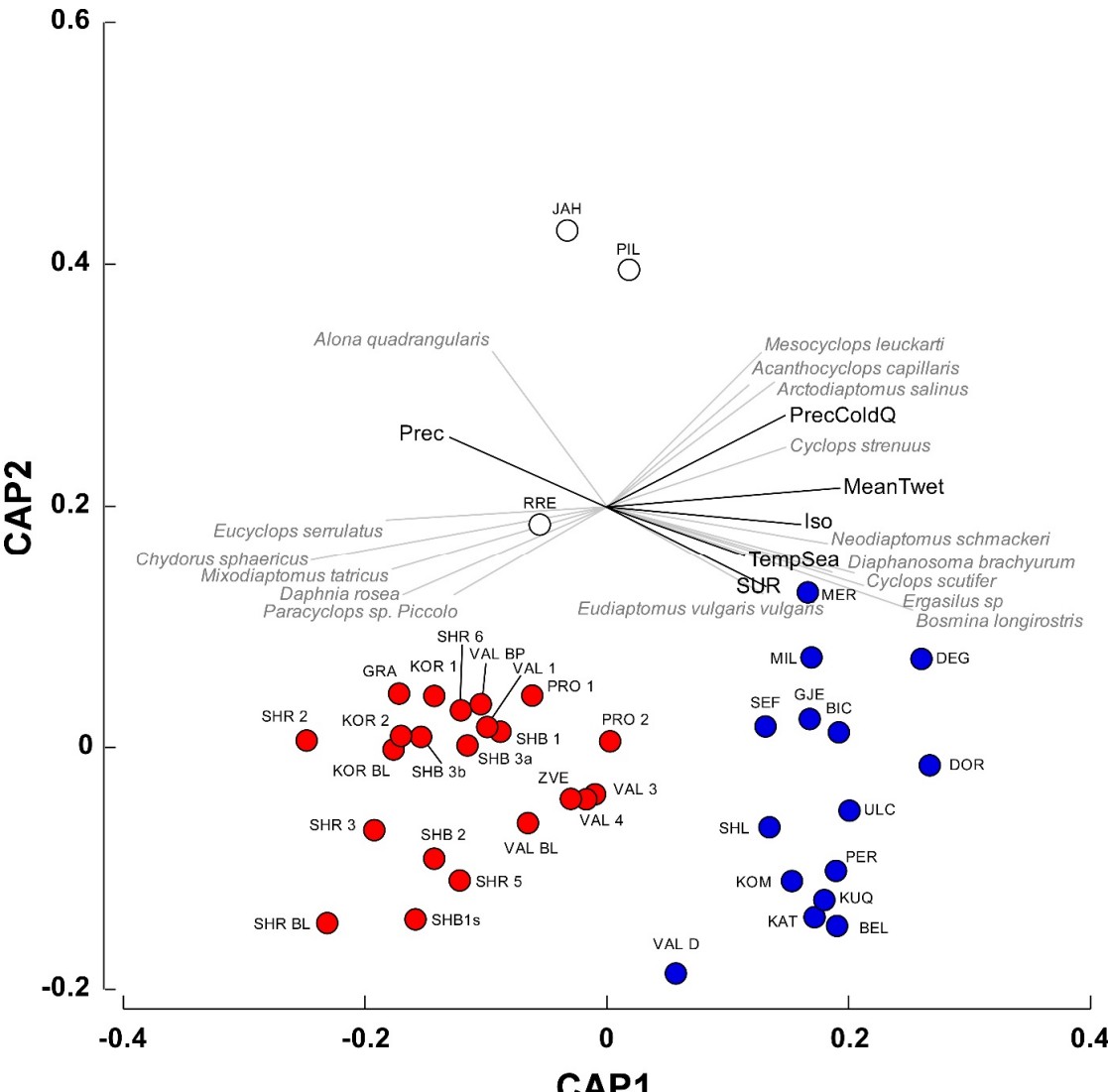

**Figure 4.** Canonical analysis of principal components (CAP) analysis (*m* = 6, misclassification = 13%, *P* = 0.0001) testing the differences in crustacean zooplankton assemblages across the 40 waterbodies included in the study as affected by bioclimatic factors. *Markers* represent waterbodies labelled with the respective code (Table 1); their color categorizes the two groups (i.e., red for group1 and blue for group2) showing significant differences in species composition (post hoc PERMANOVA, P(perm) < 0.001). Vector overlay are Spearman correlations of bioclimatic factors and species with canonical axes with *r* > 0.55.

In group1, the Mean Temperature of Wettest Quarter was remarkably lower than that determined for group2 (0.06 ± 0.69 vs. 8.44 ± 0.09 °C; *t*-test for separate variances: *t* = 9.41, *P* < 0.0001, 25.53 d.f.). Similarly, Isothermality showed lower values in group1 (33.91 ± 0.23 vs. 38.42 ± 0.24, *t* = 9.19, *P* < 0.0001, 29.52 d.f.), while the Annual Precipitation showed an inverse pattern (1098 ± 10.42 vs. 1012 ± 1.71 mm, *t* = −6.43, *P* < 0.001, 26.09 d.f.).

The analysis of the relationships between species occurrences and the canonical axis 1 (see Table A2 for Spearman correlations for the complete list of species) indicated that *Eucyclops serrulatus*, *Chydorus sphaericus*, *Mixodiaptomus tatricus*, and *Daphnia rosea* in the first group were correlated mainly with Precipitation. The occurrences of *Neodiaptomus schmackeri*, *Diapahnosoma brachyurum*, *Cyclops scutifer*, *Ergasilus* sp, *Bosmina longirostris*, and, to a lesser extent, *Mesocyclops leuckarti* in the group2 were

related with the variables Isothermality, Mean Temperature of Wettest Quarter, and Precipitation of Coldest Quarter.

## 4. Discussion

The earth is undergoing an accelerated rate of native ecosystem conversion and degradation and there is increased interest in measuring and modelling biodiversity while using landscape-scale, remotely-sensed predictors. Here, we made an attempt towards this direction while using crustacean zooplankton. This group of organisms, ubiquitous in lentic habitats, has been recently subjected to renewed interest as an effective bio-indicator of the environmental status of ponds and small lakes [62–65]. The heuristic procedure was used here to identify minimum adequate models predicting the diversity the crustacean zooplankton assemblages across the 40 waterbodies under analysis provided non-univocal results. On one hand, they showed that a subset of bioclimatic variables could effectively predict the variation in species richness and composition across the different waterbodies. On the other hand, they also indicated that the assemblages' taxonomic distinctness Δ+ is unrelated with landscape-scale environmental drivers.

Before discussing these results, it must be considered that the emphasis we put on landscape and bioclimatic drivers of zooplankton diversity by no means imply that other chemical, physical, and biotic characteristics of the waterbodies, such as nutrient concentration, pH, depth, predators, aquatic vegetation, etc. are to be considered of secondary importance. A number of studies have unequivocally indicated that they can directly affect zooplankton species richness ([66,67] and references cited in the introduction; but see also further in this section). In addition, lake age [68,69], connectivity, and, in general, the spatial arrangement of the habitat have been acknowledged to play an important structuring role (e.g., [70]; see also [71] for a marine example). However, in the present study, a hierarchy of effects at different spatial and environmental scales is implicitly assumed, with landscape and bioclimatic drivers indirectly affecting the characteristics of the biota (including crustacean zooplankton) by affecting the chemical and physical conditions of the waterbodies. Indeed, the limited extension of the basins that were included in our study (Figure 1) actually implies for them a low thermal inertia, and thus the ability to rapidly respond to the external climatic conditions [72]. An indirect support to this view is also provided by the negligible predictive power of waterbodies' area for assemblages' S, Δ+, and species composition, confirming the results of investigations performed on the benthic fauna of high-altitude ponds [22].

The best MAM included as predictors the degree of non-arboreal vegetation cover of the land areas neighboring the waterbodies, temperature seasonality, and mean temperature of the wettest quarter. The positive influence of the non-arboreal vegetation cover on species richness is consistent with the results of studies that were focused on macrobenthos in lotic habitats (e.g., [73] and literature cited). This could be ascribed to a positive, indirect effect of lateral trophic enrichment on aquatic primary producers, increasing zooplankton diversity by a phytoplankton-mediated bottom up effect or, alternatively, promoting habitat heterogeneity through an increase in aquatic vegetation [66,74,75].

Noticeably, the two temperature variables had contrasting effects on species richness: the lowest number of species was predicted to occur in waterbodies that were subjected to minimum temperature variability during the year and to maximum temperatures during the wettest months, i.e., in winter. The positive influence of temperature variability on patterns of species diversity has been acknowledged for zooplankton and other freshwater invertebrates [76,77], and can generally be ascribed to an effect of habitats environmental heterogeneity on empty niches availability, and, in turn, on species richness [78]. The negative influence of the mean temperature of the wettest quarter (MeanTwet) on species richness can be explained while considering that the former scales negatively with the altitude of the water bodies ($r = -0.95$, $P < 0.001$, 38 d.f.). Thus, MeanTwet maximum values were observed for basins that were located at low altitudes, generally in highly anthropized areas. Accordingly, the lower species richness characterizing these environments might be actually determined by the interplay of a spectrum of anthropogenic perturbations, such as pollution, water caption, or introduction of

predatory fish. Indeed, until 1990, the vast majority of low-altitudes waterbodies in Albania have been stocked with both native and non-indigenous fish species [79], and fish predation has been repeatedly recognized to influence the species richness of zooplankton in lentic habitats [80,81].

The "assemble first, predict later" modelling strategy that was used in the present study was successful in predicting species richness and is generally acknowledged to have several advantages, among others an enhanced capacity to synthesize complex data into a form more readily interpretable by scientists and decision-makers [82]. However, it is apparent that it presents important limitations, as testified by the failure in modeling taxonomic distinctness. Δ+ resulted in being not predictable, confirming the results of several investigations that have found weak or negligible relationships of the taxonomic distinctness of macrobenthic communities with environmental factors [27,83,84]. Species richness S and taxonomic distinctness Δ+ are not conceptually (or mechanistically) related, and they behave differently [44]. The lack of congruence between S and Δ+ and the negligible predictability of the latter is because S is likely to respond to short-term environmental changes in the waterbodies under analysis, while Δ+ is a proxy for phylogenetic diversity, reflecting a complex set of intrinsic and extrinsic traits and expressing evolutionary long-term adaptations to local environmental conditions [85]. The canonical analysis of principal coordinates allowed for us to partially overcome these limitations, applying an "assemble and predict together" strategy [82] in order to model changes in the species composition of planktonic assemblages and provide an advanced resolution of species-specific relationships with bioclimatic factors. The CAP analysis (Figure 4) distinguished two distinct group of waterbodies, showing different climatic characteristics in terms of isothermality, mean temperature of the wettest quarter, and annual precipitation. The first (red circles in Figure 4) was mainly constituted by high-altitude ponds and lakes (elevation 1725.3 ± 123.7 m, mean ± 1SE) distributed throughout the study area characterized by Crustacea species (e.g., *Eucyclops serrulatus*, *Mixodiaptomus tatricus*) that are peculiar of pristine alpine environments [86]. The second group (blue circles in Figure 4) comprised low-altitude karst waterbodies that were located in the Dumre area (Figure 1; elevation 239.9 ± 86.2 m, mean ± 1SE), where they are subjected to several anthropogenic pressures, including agricultural and urban pollution, eutrophication, and the introduction of non-indigenous fish species [87]. Accordingly, the group is characterized by the occurrence of *Neodiaptomus schmackeri*, an Australasian species of Chinese origin that was recently introduced in Albanian lentic habitats through fish stocking [88] and the copepod *Ergasilus* sp., whose adult females are ectoparasites of fish [89].

Regarding the three isolated waterbodies in Figure 4 (i.e., JAH, PIL, and RRE), they are artificial reservoirs located at 1330, 701, and 280 m a.s.l., respectively, being heavily affected by cattle frequentation (Belmonte, personal observation). Their isolation in the CAP diagram is due to the low species richness (three species in all the waterbodies), that might be ascribed to cattle-induced eutrophication conditions [90]. However, it is worth noting that copepods vary their body size (at the community level and even for single species) inversely with the eutrophication level [91,92]. Thus, by using a plankton net with a mesh size of 200 µm, we may have underestimated the smaller component of the planktonic assemblages thus biasing the species count.

A final consideration deserves a brief mention. In this study, the spatial resolution of the bioclimatic layers (approximately 0.64 km$^2$) was lower than the area characterizing most of the ponds and lakes included in the analysis (Figure 2). In other words, the bioclimate spatial grid "matched" the dimensions of the waterbodies, the latter being completely included (and described in terms of climate and vegetation cover) within the same grid cell. Additional studies including a wider size range of waterbodies as well as bioclimatic layers resolved at different spatial resolutions are necessary to provide a more complete picture of the actual relationships linking bioclimatic factors and the diversity of lentic zooplankton at multiple regional and environmental scales.

**Author Contributions:** Conceptualization, G.B. and G.M.; methodology, G.B., S.M. and G.M.; software, G.M.; validation, G.B.; formal analysis, G.B. and G.M.; investigation, G.B. and S.M.; resources, G.B. and S.M.; data curation, G.M.; writing—original draft preparation, G.B. S.M. and G.M.; visualization, G.M.; supervision, G.B.

**Funding:** This research received no external funding.

**Acknowledgments:** Many collaborators allowed, during 12 years, the collection of samples. Among them we are particularly grateful to Bilal Shkurtaj (Univ. of Vlore, Albania), and Giuseppe Alfonso (Univ. of Salento, Italy). The authors finally thank two anonymous reviewers whose comments greatly improved the original manuscript.

**Conflicts of Interest:** The authors declare no conflict of interest.

## Appendix A

**Table A1.** List of the 25 bioclimatic variables used in the study. Layers coded MeanT- PrecColdQ were obtained from the Wordclim v2 dataset [41] available at http://www.worldclim.org/, and refer to average monthly climate data relative to the period 1970-2000 with a 30 arc-second spatial resolution (approx. $0.92 \times 0.70$ km within the study area). The vegetation layers %tr and %nontr (% tree cover and % non-tree cover) were extracted from the MODIS Vegetation Continuous Field (VCF) product (https://lpdaac.usgs.gov/products/mod44bv006/; [93]). In the table, a third VCF variable—% bare soil (%bare, in italics)—is included, as it was used together with variables %tr and %nontr only for the estimation of habitat heterogeneity (see text for further details). The R packages *raster*, *rgdal*, and *maptools* were used for the manipulation of bioclimatic layers [94–96].

| ID# | Environmental Layer | Code |
|-----|---------------------|------|
| 1 | Annual Mean Temperature | MeanT |
| 2 | Mean Diurnal Range | MeanDrange |
| 3 | Isothermality | Iso |
| 4 | Temperature Seasonality | TempSea |
| 5 | Max Temperature of Warmest Month | MaxTwarm |
| 6 | Min Temperature of Coldest Month | MinTcold |
| 7 | Temperature Annual Range | TempArange |
| 8 | Mean Temperature of Wettest Quarter | MeanTwet |
| 9 | Mean Temperature of Driest Quarter | MeanTdry |
| 10 | Mean Temperature of Warmest Quarter | MeanTwarm |
| 11 | Mean Temperature of Coldest Quarter | MeanTcold |
| 12 | Annual Precipitation | Prec |
| 13 | Precipitation of Wettest Month | PrecWetM |
| 14 | Precipitation of Driest Month | PrecDryM |
| 15 | Precipitation Seasonality | PrecSea |
| 16 | Precipitation of Wettest Quarter | PrecWetQ |
| 17 | Precipitation of Driest Quarter | PrecDryQ |
| 18 | Precipitation of Warmest Quarter | PrecWarmQ |
| 19 | Precipitation of Coldest Quarter | PrecColdQ |
| 20 | Percent tree cover | %tr |
| 21 | Percent non-tree cover | %nontr |
|  | *Percent bare soil* | *%bare* |
| 22 | Perimeter | PER |
| 23 | Surface | SUR |
| 24 | Altitude | ELE |
| 25 | Habitat heterogeneity | H |

**Table A2.** List of the 79 crustacean species sampled in the 40 waterbodies under analysis. Linnaean classification of species using eight taxonomic levels (i.e., class, subclass, order, suborder, infraorder, family, genus and species) and the total number of occurrences are included.

| Taxon | Occurrences |
|---|---|
| **Branchiopoda** | |
| **Phyllopoda** | |
| Diplostraca | |
| **Cladocera** | |
| Anomopoda | |
| **Bosminidae** | |
| *Bosmina longirostris* | 16 |
| **Chydoridae** | |
| *Alona quadrangularis* | 8 |
| *Alona rustica* | 4 |
| *Alonella pulchella* | 1 |
| *Alonella* sp | 1 |
| *Biapertura affinis* | 3 |
| *Chydorus piger* | 2 |
| *Chydorus* sp2 | 1 |
| *Chydorus sphaericus* | 15 |
| *Coronatella rectangula* | 1 |
| *Disparalona leei* | 1 |
| *Disparalona rostrata* | 3 |
| *Eurycercus* sp | 2 |
| *Paralona pigra* | 1 |
| *Pleuroxus* sp2 | 4 |
| *Pleuroxus truncatus* | 2 |
| **Daphniidae** | |
| *Ceriodaphnia pulchella* | 1 |
| *Ceriodaphnia quadrangula* | 2 |
| *Ceriodaphnia reticulata* | 2 |
| *Ceriodaphnia setosa* | 1 |
| *Daphnia cucullata* | 2 |
| *Daphnia curvirostris* | 2 |
| *Daphnia dentifera* | 1 |
| *Daphnia galeata* | 1 |
| *Daphnia hyalina* | 8 |
| *Daphnia longispina* | 13 |
| *Daphnia pulex* | 2 |
| *Daphnia rosea* | 5 |
| *Daphnia* sp | 1 |
| *Scapholeberis kingii* | 1 |
| *Simocephalus serrulatus* | 1 |
| *Simocephalus vetulus* | 2 |
| **Ilyocryptidae** | |
| *Ilyocryptus* sp | 2 |
| **Macrothricidae** | |
| *Macrothrix* sp | 2 |
| **Moinidae** | |
| *Moina affinis* | 1 |
| *Moina brachiata* | 3 |
| *Moina macrocopa* | 2 |
| *Moina micrura* | 3 |
| Ctenopoda | 15 |
| **Sididae** | |
| *Diaphanosoma brachyurum* | 12 |
| *Diaphanosoma lacustris* | 3 |

**Table A2.** *Cont.*

| Taxon | Occurrences |
| --- | --- |
| Haplopoda | 3 |
| **Leptodoridae** | |
| *Leptodora kindtii* | 3 |
| **Hexanauplia** | |
| **Copepoda** | |
| Neocopepoda | |
| **Gymnoplea** | |
| Calanoida | |
| **Diaptomidae** | |
| *Arctodiaptomus salinus* | 4 |
| *Eudiaptomus gracilioides* | 1 |
| *Eudiaptomus vulgaris vulgaris* | 4 |
| *Eudiaptomus zachariasi* | 2 |
| *Mixodiaptomus* sp2 | 2 |
| *Mixodiaptomus tatricus* | 12 |
| *Neodiaptomus schmackeri* | 7 |
| **Podoplea** | |
| Cyclopoida | |
| **Cyclopidae** | |
| *Acanthocyclops capillaris* | 4 |
| *Acanthocyclops* sp. small | 3 |
| *Acanthocyclops trajani* | 1 |
| *Acanthocyclops vernalis* | 4 |
| *Cyclops abyssorum* | 1 |
| *Cyclops bohater* | 1 |
| *Cyclops ricae* | 1 |
| *Cyclops scutifer* | 7 |
| *Cyclops* sp8 | 1 |
| *Cyclops strenuous* | 4 |
| *Cyclops vicinus* | 1 |
| *Diacyclops bicuspidatus* | 5 |
| *Diacyclops languidoides* | 1 |
| *Eucyclops serrulatus* | 8 |
| *Halicyclops* sp | 1 |
| *Macrocyclops distinctus* | 2 |
| *Macrocyclops fuscus* | 2 |
| *Megacyclops brachypus* | 1 |
| *Mesocyclops gracilis* | 1 |
| *Mesocyclops leuckarti* | 12 |
| *Mesocyclops* sp3 | 2 |
| *Metacyclops stammeri* | 2 |
| *Microcyclops* sp | 4 |
| *Paracyclops affinis* | 2 |
| *Paracyclops cf. ectocyclops* | 1 |
| *Paracyclops fimbriatus* | 3 |
| *Paracyclops* sp. small | 3 |
| *Thermocyclops* sp | 1 |
| *Tropocyclops* sp | 4 |
| **Lernaeidae** | |
| *Lernaea* sp | 2 |
| Poecilostomatoida | 7 |
| **Ergasilidae** | |
| *Ergasilus* sp | 7 |

**Table A3.** The nine bioclimatic variables characterized by a VIF factor < 10 used for MAM analysis. For predictor abbreviations see Table A1. Iso and TempSea are expressed in dimensionless units, MeanTwet in °C, Prec and PrecColdQ in mm, %tr and %nontr in percent, SUR in ha, ELE in m, and H (referring to habitat heterogeneity estimated by the Shannon's diversity index, see text) in dimensionless units. For each variable, the among-location coefficient of variation CV and the VIF value are included.

| Waterbody | Iso | TempSea | MeanTwet | Prec | PrecColdQ | %tr | %nontr | SUR | H |
|---|---|---|---|---|---|---|---|---|---|
| BEL | 38.82 | 685.85 | 8.62 | 1023 | 334 | 1 | 20 | 25.82 | 0.241 |
| BIC | 39.19 | 668.66 | 8.5 | 1023 | 336 | 7.32 | 72.94 | 3.31 | 0.322 |
| DEG | 38.51 | 669.75 | 8.9 | 1001 | 331 | 70 | 20 | 33.07 | 0.348 |
| DOR | 38.66 | 685.47 | 8.68 | 1013 | 333 | 4.51 | 59.3 | 10.85 | 0.355 |
| GJE | 38.83 | 681.62 | 8.62 | 1018 | 333 | 6.96 | 65.85 | 2.57 | 0.354 |
| GRA | 35.08 | 621.71 | −3.1 | 1071 | 323 | 5.36 | 69.24 | 0.69 | 0.33 |
| JAH | 32.83 | 632.66 | 6.77 | 1247 | 475 | 9.64 | 58.52 | 0.21 | 0.392 |
| KAT | 39.35 | 668.39 | 8.48 | 1022 | 334 | 9.73 | 67.54 | 2.92 | 0.36 |
| KOM | 39.22 | 672.28 | 8.5 | 1019 | 332 | 3.46 | 68.54 | 1.77 | 0.318 |
| KOR 1 | 34.34 | 660.73 | −2.73 | 1077 | 305 | 9.62 | 74.95 | 0.28 | 0.317 |
| KOR 2 | 34.34 | 660.73 | −2.73 | 1077 | 305 | 9.28 | 73.14 | 0.26 | 0.328 |
| KOR BL | 36.85 | 689.46 | −1.1 | 1069 | 315 | 45.29 | 36.44 | 0.25 | 0.45 |
| KUQ | 38.51 | 678.82 | 8.78 | 1008 | 331 | 8.21 | 72.68 | 6.53 | 0.327 |
| MER | 38.76 | 685.2 | 8.77 | 1008 | 333 | 2 | 50 | 59.9 | 0.337 |
| MIL | 39.19 | 668.66 | 8.5 | 1023 | 336 | 10.75 | 66.78 | 5.85 | 0.367 |
| PER | 38.71 | 667.18 | 9.03 | 1001 | 334 | 9 | 42 | 26.08 | 0.404 |
| PIL | 35.16 | 621.14 | 8.93 | 1281 | 510 | 8.46 | 70.5 | 0.44 | 0.34 |
| PRO 1 | 33.33 | 631.13 | 3.73 | 1098 | 343 | 3.46 | 57.38 | 0.0091 | 0.348 |
| PRO 2 | 32.91 | 635.5 | 3.83 | 1089 | 336 | 19.42 | 55.96 | 0.0148 | 0.429 |
| RRE | 29.43 | 609.67 | 4.82 | 1116 | 349 | 5.39 | 52.62 | 0.0136 | 0.373 |
| SEF | 38.84 | 682.17 | 8.65 | 1013 | 333 | 5 | 35 | 85.87 | 0.358 |
| SHB 1 | 34.7 | 650.77 | −1.27 | 1091 | 322 | 11.2 | 71.83 | 0.38 | 0.34 |
| SHB 2 | 33.75 | 629.99 | −2.13 | 1121 | 323 | 10.7 | 71.3 | 1.44 | 0.343 |
| SHB 3a | 33.53 | 623.2 | −2.42 | 1126 | 325 | 9.06 | 68.15 | 1.53 | 0.354 |
| SHB 3b | 33.53 | 623.2 | −2.42 | 1126 | 325 | 10.66 | 67.87 | 1.17 | 0.361 |
| SHB1s | 34.11 | 630.07 | −1.98 | 1114 | 323 | 11.2 | 71.83 | 0.0145 | 0.34 |
| SHL | 36.98 | 669.26 | 7.45 | 989 | 277 | 30 | 50 | 14.96 | 0.447 |
| SHR 2 | 32.68 | 643.99 | −4.73 | 1077 | 283 | 4.85 | 80.93 | 1.35 | 0.259 |
| SHR 3 | 33.4 | 666.58 | −0.53 | 1056 | 277 | 7.26 | 78.48 | 6.02 | 0.286 |
| SHR 5 | 32.2 | 615.66 | −5.32 | 1107 | 290 | 3.96 | 72.99 | 0.45 | 0.302 |
| SHR 6 | 33.23 | 631.91 | −4.05 | 1057 | 275 | 8.53 | 79.14 | 1.74 | 0.284 |
| SHR BL | 32.48 | 635.84 | −4.7 | 1083 | 282 | 5.43 | 79.98 | 3.26 | 0.268 |
| ULC | 39.35 | 668.39 | 8.48 | 1022 | 334 | 8.29 | 63.82 | 6.46 | 0.369 |
| VAL 1 | 35.34 | 615.94 | 1.23 | 1099 | 324 | 6.96 | 73.44 | 0.72 | 0.318 |
| VAL 3 | 35.12 | 613.86 | −2.25 | 1108 | 328 | 6.32 | 69.34 | 0.38 | 0.335 |
| VAL 4 | 35.12 | 614.42 | −2.03 | 1108 | 327 | 8.09 | 65.35 | 2.38 | 0.362 |
| VAL BL | 35.66 | 636.2 | 3.47 | 1066 | 298 | 14.58 | 68.43 | 2.43 | 0.365 |
| VAL BP | 35.54 | 617.85 | 1.35 | 1103 | 326 | 9.2 | 75.11 | 1.61 | 0.315 |
| VAL D | 33.4 | 671.99 | 6.63 | 999 | 311 | 27.69 | 45.71 | 34.74 | 0.463 |
| ZVE | 33.16 | 583.76 | 11 | 903 | 356 | 10 | 30 | 0.29 | 0.39 |
| CV | 7.47 | 4.24 | 167.34 | 6.31 | 13.12 | 108.96 | 25.83 | 202.81 | 13.92 |
| VIF | 2.62 | 1.84 | 6.75 | 5.15 | 6.17 | 1.91 | 2.01 | 4.72 | 1.81 |

**Table A4.** Spearman rank correlation coefficients of crustacean zooplankton species with the six axes (CAP1- CAP6) extracted by canonical analysis of principal coordinates. Coefficients > 0.55 (in absolute terms) are in bold.

| Taxon | CAP1 | CAP2 | CAP3 | CAP4 | CAP5 | CAP6 |
|---|---|---|---|---|---|---|
| *Acanthocyclops capillaris* | 0.36 | 0.31 | −0.1 | −0.01 | −0.27 | −0.28 |
| *Acanthocyclops* sp. small | −0.37 | 0 | 0.03 | −0.25 | 0 | 0 |
| *Acanthocyclops trajani* | −0.22 | 0.19 | −0.17 | −0.24 | −0.03 | −0.03 |
| *Acanthocyclops vernalis* | 0.36 | −0.06 | 0.08 | −0.13 | 0.08 | 0.34 |
| *Alona quadrangularis* | −0.29 | 0.4 | −0.23 | −0.37 | 0.19 | −0.02 |
| *Alona rustica* | −0.03 | −0.16 | −0.29 | 0.47 | −0.12 | 0.38 |
| *Alonella pulchella* | 0.08 | −0.27 | −0.1 | 0.22 | 0.01 | 0.24 |
| *Alonella* sp | −0.15 | 0.16 | −0.13 | −0.09 | 0.03 | 0.16 |
| *Arctodiaptomus salinus* | 0.43 | 0.32 | −0.19 | −0.03 | −0.25 | 0.03 |
| *Biapertura affinis* | −0.23 | −0.08 | −0.11 | −0.04 | −0.03 | 0.03 |
| *Bosmina longirostris* | **0.78** | −0.27 | 0.08 | −0.18 | 0.17 | 0 |
| *Ceriodaphnia pulchella* | 0.16 | 0.22 | −0.09 | −0.01 | −0.09 | 0.13 |
| *Ceriodaphnia quadrangula* | 0.26 | 0.21 | 0.11 | 0.12 | −0.29 | −0.17 |
| *Ceriodaphnia reticulata* | −0.17 | 0.16 | 0.29 | −0.13 | −0.28 | 0.14 |
| *Ceriodaphnia setosa* | 0.16 | 0.22 | −0.09 | −0.01 | −0.09 | 0.13 |
| *Ceriodaphnia* sp3 | 0.17 | −0.22 | 0.19 | 0.02 | 0.1 | −0.24 |
| *Chydorus piger* | −0.04 | 0.03 | 0.18 | 0.36 | 0.08 | −0.02 |
| *Chydorus* sp2 | −0.09 | 0.15 | 0.08 | 0.19 | 0.22 | 0.23 |
| *Chydorus sphaericus* | **−0.75** | −0.14 | −0.44 | −0.34 | −0.23 | −0.2 |
| *Coronatella rectangula* | −0.23 | −0.02 | 0.22 | −0.13 | −0.05 | −0.16 |
| *Cyclops abyssorum* | 0.01 | −0.08 | −0.26 | 0.27 | −0.2 | 0.19 |
| *Cyclops bohater* | −0.08 | 0.09 | 0.23 | 0.24 | 0.24 | 0.03 |
| *Cyclops ricae* | 0.23 | 0.06 | 0.05 | 0.12 | −0.19 | −0.26 |
| *Cyclops scutifer* | **0.57** | −0.17 | 0.09 | −0.12 | 0.11 | −0.24 |
| *Cyclops* sp8 | −0.12 | 0.13 | 0.16 | 0.15 | 0.2 | −0.08 |
| *Cyclops strenuous* | 0.45 | 0.15 | −0.01 | −0.03 | −0.17 | −0.38 |
| *Cyclops vicinus* | 0.15 | 0.12 | 0.03 | −0.22 | 0.09 | 0.01 |
| *Daphnia cucullata* | 0.16 | 0.24 | −0.09 | −0.09 | −0.27 | 0.12 |
| *Daphnia curvirostris* | −0.04 | 0.03 | 0.18 | 0.36 | 0.08 | −0.02 |
| *Daphnia dentifera* | −0.15 | 0.16 | −0.13 | −0.09 | 0.03 | 0.16 |
| *Daphnia galeata* | 0.16 | 0.22 | −0.09 | −0.01 | −0.09 | 0.13 |
| *Daphnia hyalina* | −0.3 | 0.03 | −0.39 | 0.44 | −0.19 | 0.01 |
| *Daphnia longispina* | −0.24 | 0.05 | 0.13 | **0.65** | 0 | −0.35 |
| *Daphnia pulex* | 0.01 | 0.13 | 0.38 | −0.02 | −0.38 | 0.24 |
| *Daphnia rosea* | −0.52 | −0.23 | −0.13 | −0.54 | 0.08 | −0.11 |
| *Diacyclops bicuspidatus* | −0.13 | 0.21 | 0.46 | −0.14 | −0.43 | 0.13 |
| *Diacyclops languidoides* | −0.12 | 0.13 | 0.16 | 0.15 | 0.2 | −0.08 |
| *Diaphanosoma brachyurum* | **0.63** | −0.17 | 0.21 | −0.28 | 0.1 | 0.36 |
| *Diaphanosoma lacustris* | 0.22 | 0.27 | −0.05 | −0.2 | −0.17 | 0.1 |
| *Disparalona leei* | 0.19 | −0.2 | 0.06 | −0.16 | 0.15 | 0.26 |
| *Disparalona rostrata* | −0.37 | −0.14 | 0.03 | −0.17 | 0.16 | −0.08 |
| *Ergasilus* sp | **0.65** | −0.2 | 0.03 | −0.07 | −0.01 | −0.2 |
| *Eucyclops serrulatus* | **−0.56** | −0.03 | 0.03 | −0.3 | 0.25 | −0.15 |
| *Eudiaptomus gracilis* | 0.08 | −0.27 | −0.1 | 0.22 | 0.01 | 0.24 |
| *Eudiaptomus vulgaris vulgaris* | 0.4 | −0.22 | 0.29 | 0.16 | 0.01 | −0.52 |
| *Eudiaptomus zachariasi* | −0.25 | 0.15 | 0 | −0.2 | −0.06 | 0.17 |
| *Eurycercus* sp | 0.29 | −0.33 | 0.05 | −0.22 | 0.19 | 0.33 |
| *Halicyclops* sp | −0.2 | 0.05 | 0.13 | −0.19 | −0.12 | 0.08 |
| *Ilyocryptus* sp | −0.05 | −0.03 | −0.27 | −0.01 | −0.19 | 0.01 |
| *Leptodora kindtii* | 0.15 | −0.18 | −0.04 | −0.23 | 0.18 | 0.37 |
| *Lernaea* sp | 0.34 | −0.14 | −0.05 | −0.14 | 0.13 | 0.2 |

**Table A4.** *Cont.*

| Taxon | CAP1 | CAP2 | CAP3 | CAP4 | CAP5 | CAP6 |
|---|---|---|---|---|---|---|
| *Macrocyclops distinctus* | −0.25 | 0.15 | 0 | −0.2 | −0.06 | 0.17 |
| *Macrocyclops fuscus* | −0.22 | −0.04 | −0.2 | −0.02 | −0.15 | −0.09 |
| *Macrothrix* sp | −0.36 | −0.27 | −0.08 | −0.31 | 0.05 | −0.04 |
| *Mesocyclops gracilis* | 0.09 | 0.1 | −0.23 | −0.17 | −0.16 | 0.15 |
| *Mesocyclops leuckarti* | 0.39 | 0.39 | −0.52 | 0.26 | −0.52 | 0 |
| *Mesocyclops* sp3 | −0.2 | −0.06 | 0.01 | 0.1 | 0.05 | 0.2 |
| *Metacyclops stammeri* | 0.04 | −0.06 | 0.25 | 0.12 | 0.22 | −0.23 |
| *Microcyclops* sp | 0.03 | −0.26 | −0.32 | 0.51 | −0.3 | 0.22 |
| *Mixodiaptomus* sp2 | −0.2 | 0.02 | −0.02 | 0.15 | 0.03 | −0.26 |
| *Mixodiaptomus tatricus* | **−0.55** | −0.16 | −0.35 | 0.17 | −0.17 | −0.09 |
| *Moina affinis* | −0.19 | −0.23 | −0.06 | −0.05 | −0.15 | 0.05 |
| *Moina brachiata* | 0.29 | −0.05 | 0.11 | 0.26 | −0.23 | −0.35 |
| *Moina macrocopa* | 0.01 | 0.13 | 0.38 | −0.02 | −0.38 | 0.24 |
| *Moina micrura* | 0.17 | −0.02 | 0.28 | 0.01 | 0.41 | 0.25 |
| *Neodiaptomus schmackeri* | **0.56** | −0.09 | −0.17 | −0.37 | 0.08 | 0.28 |
| *Paracyclops affinis* | −0.25 | 0.15 | 0 | −0.2 | −0.06 | 0.17 |
| *Paracyclops cf. ectocyclops* | −0.17 | 0.03 | −0.01 | 0.08 | 0.06 | −0.19 |
| *Paracyclops fimbriatus* | −0.17 | 0.22 | 0.28 | 0.34 | 0.39 | 0.11 |
| *Paracyclops* sp. small | −0.39 | −0.23 | 0.07 | −0.23 | −0.1 | −0.07 |
| *Paralona pigra* | 0.16 | 0.22 | −0.09 | −0.01 | −0.09 | 0.13 |
| *Pleuroxus* sp2 | −0.22 | 0.14 | −0.09 | 0.09 | −0.04 | 0.34 |
| *Pleuroxus truncatus* | −0.12 | −0.29 | −0.1 | 0.01 | 0.02 | 0.17 |
| *Scapholeberis kingii* | −0.2 | 0.05 | 0.13 | −0.19 | −0.12 | 0.08 |
| *Simocephalus serrulatus* | −0.18 | −0.14 | 0.1 | −0.08 | 0.2 | −0.16 |
| *Simocephalus vetulus* | −0.1 | 0.06 | −0.28 | 0.13 | −0.12 | 0.25 |
| *Thermocyclops* sp | 0.12 | −0.2 | 0.26 | 0.08 | 0.26 | −0.33 |
| *Tropocyclops* sp | −0.08 | 0.39 | 0.26 | 0.35 | 0.47 | 0.17 |

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
