# Peer review of "Species Richness and Taxonomic Distinctness of Zooplankton in Ponds and Small Lakes from Albania and North Macedonia: The Role of Bioclimatic Factors"

_water, doi:10.3390/w11112384_

Round 1

Reviewer 1 Report

In this manuscript the authors studied the species richness and taxonomic distinctness of crustacean zooplankton assemblages from several ponds and lakes in relation to some bioclimatic and morphometric variables. The manuscript is overall well-written and presents some relevant results that will be of interested by a specialized audience. The use of data derived from Wordclim and MODIS databases is interesting and quite novel in zooplankton studies.

However, there are some issues that need to be corrected or better explained.

The most important is that the authors need to make clear (particularly when discussing the results) that the study only focuses on the influence of some bioclimatic variables whereas several other local and regional environmental and biotic variables (such as nutrient concentration, pH, predators, depth, connectivity, etc.) are not taken into account, although there are several studies that show that are important at predicting zooplankton species richness. Authors need to discuss how this may influence their results.

The title is no precise enough. The study focus on species richness, not on species diversity.

The information presented in the Abstract is a bit weird. There is more focus on the variables that do not explain any pattern that in variables that do. Please explain which variables influenced species richness and how.

I think that Wordclim and MODIS databases can offer interesting data for biodiversity studies. However, it would be interesting to discuss how the resolution chosen can influence the results because of the different sizes of the ponds and lakes.

Specific comments:

PL36-39. Please see a more recent review supporting this statement for Europe: Gozlan et al 2019, Inland Waters, 9, 78-94.

L117-125: Potential differences in the results due to different sample sizes can be overcome by using rarefaction curves. Rarefaction curves resample from a pooled group of samples to plot the expected number of species against an increasing number of samples, thus adjusting for differences in sample size.

L119. The word “interested” does not seem to fit here.

L136-137. “Measurements were performed by preferentially choosing images taken in spring or summer between 2006 and 2017.” Why did you chose this long time range? Ponds may have changed during this time frame.

L 219 “Given the outcomes of the analyses (see Results section) we verified whether bioclimatic factors influenced planktonic Crustacea assemblages in terms of species composition”. Explain briefly the outcomes mentioned here since it is difficult to understand which is the purpose of these analyses.

Instead of calling the variables by using a code (e.g. BIO03, BIO08, etc) I think it will be better to call them by their name or by an abbreviation so that one can easily recall which variable is which.

Author Response

Lecce, 06/11/2019

All the comments and criticisms raised by the reviewers have been accepted, with only two exceptions. Given the substantial variations introduced in the reference list, tables, and figures, changes were made on the original manuscript. We also propose a variation in the Author order (not Belmonte, Mali, Mancinelli, but Mancinelli, Mali, Belmonte) confirming the corresponding author. We are now confident that the manuscript is appropriate for publication in Water.

Specifically:

REVIEWER1

The most important is that the authors need to make clear (particularly when discussing the results) that the study only focuses on the influence of some bioclimatic variables whereas several other local and regional environmental and biotic variables (such as nutrient concentration, pH, predators, depth, connectivity, etc.) are not taken into account, although there are several studies that show that are important at predicting zooplankton species richness. Authors need to discuss how this may influence their results.

Accepted. The discussion section has been partially re-written, acknowledging the comment

The title is no precise enough. The study focus on species richness, not on species diversity.

Accepted; the title has been made more explicit

The information presented in the Abstract is a bit weird. There is more focus on the variables that do not explain any pattern that in variables that do. Please explain which variables influenced species richness and how.

Accepted; the abstract was partially re-written

I think that Wordclim and MODIS databases can offer interesting data for biodiversity studies. However, it would be interesting to discuss how the resolution chosen can influence the results because of the different sizes of the ponds and lakes.

Accepted; a brief conclusive paragraph addressing the issue has been added in the discussion

PL36-39. Please see a more recent review supporting this statement for Europe: Gozlan et al 2019, Inland Waters, 9, 78-94.

Accepted; reference added

L117-125: Potential differences in the results due to different sample sizes can be overcome by using rarefaction curves. Rarefaction curves resample from a pooled group of samples to plot the expected number of species against an increasing number of samples, thus adjusting for differences in sample size.

Not accepted. In principle, we agree with the referee on the usefulness of rarefaction curves in diversity studies. However, in the context of the present investigation the main aim was to use landscape and bioclimatic variables to model species richness, taxonomic distinctness, and species composition of “true”, not “adjusted” assemblages. In addition, rarefaction curves may have been useful only for species richness, while no comparable methods, to our knowledge, are available for taxonomic distinctness, and to adjust the species composition matrix for CAP analysis. In conclusion, we are confident that the preliminary analysis performed on the data and demonstrating that no significant artefacts occurred due to different sample number is robust enough to make the obtained results reliable.

L119. The word “interested” does not seem to fit here.

Accepted; sentence partially rephrased

L136-137. “Measurements were performed by preferentially choosing images taken in spring or summer between 2006 and 2017.” Why did you chose this long time range? Ponds may have changed during this time frame.

Indeed, the sentence contained an error, as water bodies were sampled between 2008 and 2017 (Table 1). The error was corrected. Regarding the note: we chose images of the water bodies taken between 2008 and 2017, a time lag consistent with the sampling period. The underlying assumption was that the main variations in size and morphology in the water bodies may have occurred only on a seasonal basis, with negligible changes during the sampling period. A sentence was added in order to make this assumption explicit.

L 219 “Given the outcomes of the analyses (see Results section) we verified whether bioclimatic factors influenced planktonic Crustacea assemblages in terms of species composition”. Explain briefly the outcomes mentioned here since it is difficult to understand which is the purpose of these analyses.

Accepted; a brief explanation was added;

Instead of calling the variables by using a code (e.g. BIO03, BIO08, etc) I think it will be better to call them by their name or by an abbreviation so that one can easily recall which variable is which

Accepted; variables were re-named, and an effort was made to use their full length names in the text

Reviewer 2 Report

Dear authors, first of all, I would like to praise such an interesting and well-written article. The authors analyzed the role of bioclimatic factors on the diversity patterns of crustacean zooplankton in 40 small waterbodies Albania and North Macedonia. The results of this study indicated the weak effect of most bioclimatic drivers, however some of them might influence crustacean species richness and their community structures (temperature and land areas neighboring the waterbodies). My overall impression is that the statistical approach is very well planned and detailed described in methods. The results are very clearly presented and well described. The other advantage of the study is that the authors include a full list of species for this region. Whereas the Introduction, Discussion and Conclusion sections can be improved. Below I attach comments to individual chapters, which I hope may be useful to apply before publication.

(i) The introduction is too general. Maybe the authors could further substantiate why the bioclimatic factors are important for zooplankton diversity. There is also a lot of information about the size of the waterbodies (SAR), however these results were not discussed.

(ii) The first part of the discussion often applies to completely different groups of organisms (macrobenthos or even birds). It could be better to provide more details about the effect of this bioclimatic factors (VEG2, BIO4, BIO8) on the zooplankton communities. Maybe the information in this articles could be useful (DOI: 10.1007/s10750-016-2913-5; DOI: 10.1002/iroh.201301704/ab; DOI: 10.1016/j.ecolind.2018.11.025). While the second part of the discussion is well written (from 420).

(iii) Conclusions are very general and not well present the main finding of this study.

(iv) Please check the species names and their current taxonomic status, e.g. Alona rectangular – it should be Coronatella rectangula; Ceriodaphnia reticulate – it should be  Ceriodaphnia reticulata).

Finally, I think that the studies bring some new valuable information to the knowledge of limnology, therefore my suggestion is to consider the manuscript for publication after it being corrected.

Author Response

Lecce, 06/11/2019

All the comments and criticisms raised by the reviewers have been accepted, with only two exceptions. Given the substantial variations introduced in the reference list, tables, and figures, changes were made on the original manuscript. We also propose a variation in the Author order (not Belmonte, Mali, Mancinelli, but Mancinelli, Mali, Belmonte) confirming the corresponding author. We are now confident that the manuscript is appropriate for publication in Water.

Specifically:

REVIEWER2

(i) The introduction is too general. Maybe the authors could further substantiate why the bioclimatic factors are important for zooplankton diversity.

Accepted. However, for the sake of conciseness we integrated a series of considerations on the relationships between bioclimate and zooplankton diversity in the discussion section

There is also a lot of information about the size of the waterbodies (SAR), however these results were not discussed.

Not accepted. In general, the variable “area” showed no predictive power as regards with species richness, taxonomic distinctness and species composition. We briefly addressed this point in the discussion section, but, given the negligible results, we also think that it is not necessary to excessively emphasize this issue in the introduction.

(ii) The first part of the discussion often applies to completely different groups of organisms (macrobenthos or even birds). It could be better to provide more details about the effect of this bioclimatic factors (VEG2, BIO4, BIO8) on the zooplankton communities. Maybe the information in this articles could be useful (DOI: 10.1007/s10750-016-2913-5; DOI: 10.1002/iroh.201301704/ab; DOI: 10.1016/j.ecolind.2018.11.025). While the second part of the discussion is well written (from 420).

Accepted; all references suggested included with the exception of DOI: 10.1016/j.ecolind.2018.11.025, as it focuses on macrophytes and has been judged not consistent with the topic of the manuscript

(iii) Conclusions are very general and not well present the main finding of this study.

Accepted; conclusions have been rephrased and included in the discussion section

(iv) Please check the species names and their current taxonomic status, e.g. Alona rectangular – it should be Coronatella rectangula; Ceriodaphnia reticulate – it should be  Ceriodaphnia reticulata).

Accepted; current taxonomic status checked, and changes made
